# Characterization of the Immune Response to PD-1 Blockade during Chemoradiotherapy for Head and Neck Squamous Cell Carcinoma

**DOI:** 10.3390/cancers14102499

**Published:** 2022-05-19

**Authors:** Juan L. Callejas-Valera, Daniel W. Vermeer, Christopher T. Lucido, Caitlin Williamson, Marisela Killian, Paola D. Vermeer, William C. Spanos, Steven F. Powell

**Affiliations:** 1Sanford Research, 2301 East 60th Street North, Sioux Falls, SD 57104, USA; juanlu.patinalba@gmail.com (J.L.C.-V.); danielvermeer100@gmail.com (D.W.V.); caitlin.williamson@Sanfordhealth.org (C.W.); maudikillian@gmail.com (M.K.); paola.vermeer@sanfordhealth.org (P.D.V.); william.spanos@sanfordhealth.org (W.C.S.); 2Sanford School of Medicine, University of South Dakota, 1400 W 22nd Street, Sioux Falls, SD 57105, USA; chris.lucido@coyotes.usd.edu

**Keywords:** head and neck cancer, immunotherapy, PD-1, chemoradiotherapy, T cells, HNSCC

## Abstract

**Simple Summary:**

Blockade of the PD-1/L1 interaction represents a breakthrough in the treatment for recurrent/metastatic head and neck squamous cell carcinoma (HNSCC). Clinical and translational research suggests that this interaction may play a role in immune evasion during chemoradiotherapy. Using an immune-competent murine model of HNSCC, we demonstrate improved efficacy of PD-1 blockade with concurrent cisplatin-based chemoradiotherapy. Taking this approach into a clinical trial evaluating the anti-PD-1 agent, pembrolizumab, combined with chemoradiotherapy in HNSCC, we characterized the peripheral blood immune response to therapy. Our findings highlight that this combination is active in the murine model and circulating PD-1+ T-cell proportions were decreased during the clinical trial. However, additional findings from the clinical trial suggest a shift towards expression of other markers of immune exhaustion. As this treatment approach is being explored in large, randomized trials, these findings provide insight into potential pathways for treatment failure.

**Abstract:**

Background: Chemoradiotherapy is a standard treatment for HNSCC. Blockade of the PD-1/L1-2 interaction may represent a target to overcome immune escape during this treatment. Methods: Utilizing a HNSCC mEERL C57BL/6 mouse model, we evaluated a PD-1 blockade alone or in combination with cisplatin-based chemoradiotherapy. Next, we evaluated peripheral blood mononuclear cells (PBMCs) with relative PD-1, TIM-3, and LAG-3 expression, and myeloid-derived suppressor-like (MDSC-like) populations from a clinical trial evaluating PD-1 blockade with chemoradiotherapy in HNSCC. Finally, we analyzed the effect of therapy on human T-cell clonality through T-cell Receptor (TCR) sequencing. Results: Anti-PD-1 monotherapy induced no response in the mEERL model; however, combination with chemoradiotherapy improved tumor clearance and survival. PBMCs from patients treated with this combination therapy demonstrate a decline in circulating T-cell populations with knockdown of PD-1 expressing CD3+CD4+ and CD3+CD8+ T cells during treatment. However, TIM-3, LAG-3 expressing T-cell and MDSC-like populations concordantly rose. During treatment, the TCR repertoire demonstrates overall clonal expansion, with both unique and previously reported T-cell clones. Conclusions: Our murine HNSCC model demonstrates efficacy of PD-1 blockade during chemoradiotherapy. However, while PD-1-expressing T cells decreased with this therapy, human PBMC findings also identified an increase in populations contributing to immune exhaustion. These findings further characterize PD-1 blockade during chemoradiotherapy for HNSCC and highlight potential competing mechanisms of immune evasion.

## 1. Introduction

Worldwide, head and neck squamous cell carcinomas (HNSCCs) account for approximately 650,000 diagnoses and over 300,000 deaths per year [1]. Risk factors include tobacco and alcohol use [2], as well as infection with human papillomavirus (HPV) in those with oropharyngeal tumors [3]. These unique risk factors generate cancers of the head and neck that are distinct in their presentation, treatment response, and overall survival and have therefore led to a definition of risk groups based on HPV-positive (HPV+) and HPV-negative (HPV−) disease. While HPV+ patients generally have improved response to therapies and outcomes over their HPV− counterparts, they often present with advanced nodal disease at diagnosis [4]. For this reason, many patients with HNSCC present with disease that is challenging to resect surgically or requires non-surgical approaches for organ preservation. In these advanced-stage patients, definitive chemoradiotherapy has become the standard therapeutic approach [5]. The most common treatment is cisplatin-based chemoradiotherapy.

While the primary mechanism of action for chemoradiotherapy is likely due to direct cytotoxicity, the host immune response also critically contributes to treatment response. Using a murine model of HNSCC, our group previously demonstrated that this immune response is strongly dependent on an intact CD4+ and CD8+ cellular response [6]. Furthermore, radiation may induce tumoral degradation of the “self” marker CD47 which allows for increased dendritic cell uptake and subsequent interferon gamma and granzyme production from infiltrating lymphocytes [7]. Therapeutically, activation of the immune costimulatory molecule, CD137, during chemoradiotherapy can augment immune clearance in this model [8]. As a result, investigation of novel immunotherapeutic approaches during chemoradiotherapy are of great interest using this murine model.

The programmed death-1 receptor interaction with its ligands (PD-1:PD-L1/2) has emerged as a target for immunotherapy in a number of malignancies [9]. While T-cell (expressing PD-1) engagement in the host tissue environment (expressing PD-L1 and/or PD-L2) may downregulate excessive immune response, dysregulation of this system in malignancy may promote immune evasion [10,11]. In many HNSCCs, PD-L1 expression is abundant, suggesting the importance of this pathway on tumor immune evasion [12]. Recent research suggests that the immune checkpoint interaction of PD-1:PD-L1 contributes to immune resistance in HNSCC during chemoradiotherapy. Preclinical data demonstrate that radiation [13,14] and cisplatin [15,16] independently induce PD-L1 expression in tumor cells. Human data from HNSCC patients undergoing chemoradiotherapy demonstrate an overall decline in circulating CD4+ and CD8+ T-cell populations over the course of treatment, with a rise in the proportion of cells expressing PD-1 [17]. These data suggest that the PD-1:PD-L1/2 interaction may promote immune escape during this treatment.

In addition to PD-1:PD-L1/2, other immune checkpoints leading to exhaustion are emerging as mechanisms of immune escape and potential therapeutic targets. Two of these are T-cell immunoglobulin and mucin-domain containing-3 (TIM-3) and Lymphocyte activation gene-3 (LAG-3). TIM-3 is a type I transmembrane protein that has been shown to be highly expressed on tumor antigen-specific T cells in the peripheral blood and among tumor-infiltrating lymphocytes, suggesting a role in tumor immunity [18]. Upregulation of TIM-3 is associated with the exhaustion of tumor antigen-specific CD8+ T cells, which can be reversed with inhibitory monoclonal antibodies [19]. In HNSCC, elevated expression of TIM-3 has been shown to correlate with worse clinical outcomes [20]. LAG-3 is an immune checkpoint expressed on T-, B-, Natural Killer- and dendritic cells and has recently been described as a cause of immune cell exhaustion in the setting of cancer [21]. In HNSCC, higher LAG-3 expression in human specimens has correlated with higher grade tumors, larger size, positive nodes, and poorer survival. Inhibition of LAG-3 in an immune-competent model of HNSCC was able to slow tumor growth, suggesting that it may have therapeutic potential in this disease [22]. Currently, TIM-3 and LAG-3 blocking monoclonal antibodies are being evaluated in a number of clinical trials as monotherapy and in combination with anti-PD-1:PD-L1 therapies in HNSCC [23,24]. As these are emerging therapies, investigation of their role in immune escape during chemoradiotherapy is warranted.

Monoclonal antibodies that block the PD-1:PD-L1/2 axis are now clinically used to treat a variety of cancers [25]. Based on our prior observations of the importance of the immune response during chemoradiotherapy in our murine model and the potential for this pathway leading to therapeutic resistance in human disease, we sought to determine the effect of adding PD-1 blockade to standard-of-care therapy. First, we characterized tumor-infiltrating immunocyte populations in our model to define the extent of the PD-1+ T-cell infiltration. Next, we evaluated the effect of adding PD-1 blockade during cisplatin-based chemoradiotherapy. Finally, utilizing biospecimens from a first-in-human clinical trial of pembrolizumab combined with cisplatin-based chemoradiation, we analyzed the host immunocyte response to PD-1 blockade.

## 2. Materials and Methods

### 2.1. Animal Studies

Mouse oral/pharyngeal epithelial (MOE) cells expressing HPV16 E6/E7, hRas, and luciferase (mEERL) previously developed by our laboratory group [26,27,28] were used for in vivo animal studies. The mEERL cell line was maintained in DMEM supplemented with 22.5% Ham’s F12 nutrient mixture, 10% FBS, 100 U/mL penicillin, 100 μg/mL streptomycin, 0.5 μg/mL hydrocortisone, 5 μg/mL transferrin, 5 μg/mL insulin, 0.00136 μg/mL tri-iodo-thyronine, and 5 μg/mL EGF (Invitrogen). Cells were maintained in a humidified 37 °C incubator in 5% CO_2_ and screened to ensure that they were free of mycoplasma. The mEERL cell line was authenticated by short tandem repeat (STR) DNA profiling prior to use. We froze multiple vials of the first passage after STR verification. We routinely use only these early passages of cells, up to passage 15, for all in vivo experiments. In addition, this cell line has also gone through RADIL testing for bacteria, virus, and parasite contamination, including mycoplasma, as previously described [29].

All experiments were performed at the Sanford Research Animal Research Center (ARC) in accordance with Sanford Research Institutional Animal Care and Use Committee (IACUC) approval. Sanford ARC Animal Welfare Assurance is on file with the Office of Laboratory Animal Welfare under assurance number A-4568-01. Sanford is also licensed under the United States Department of Agriculture (USDA) and the Sanford ARC is Association for Assessment and Accreditation of Laboratory Animal Care (AAALAC)-accredited. For all studies, four- to six-week-old, 20–25 gm male C57BJ/6 mice were obtained from the Jackson Laboratory. Mice were maintained under pathogen-free conditions with a 14:10 light/dark cycle and ad libitum access to food and water. Tumors were initiated by injecting 10^5^ mEERL cells in 100 µL of DMEM subcutaneously at a single injection site (day 0) in the right hind flank of mice with a 23-gauge needle. Implantation success using this technique is 100%. Mice were euthanized when sacrifice criteria (tumor volume, ulceration, excessive edema, or emaciation) were met. All animals were assigned randomly to groups and investigators were blinded to treatment conditions.

For tumor-infiltrating immunocyte characterization, tumors were harvested 7, 14, 21 and 28 days post tumor implantation. Tumors were dissociated using a tumor dissociation kit (cat: 130-096-730 Miltenyi Biotech, Paris, France) and analyzed by flow cytometry evaluating T-cell populations, immune checkpoint expression, and myeloid-derived suppressor cell (MDSC) populations (see Appendix A for fluorochrome-conjugated antibody panels).

For anti-PD-1 single-agent experiments, mice (*n* = 10/group) were treated with 200 mcg of mouse anti-PD-1 antibody (DX400, Merck, San Francisco, CA, USA) administered via intraperitoneal (IP) injection after a palpable tumor developed (day 12). Treatment continued with 200 mcg of DX400 IP every 4 days for 5 treatments (day 12, day 16, day 20, day 24, day 28). Appropriate IgG1 isotype control antibodies were used in the control groups (Merck). Treatment is outlined in Figure 1B(i).

For chemoradiation experiments, after palpable tumors developed (day 12), mice (*n* = 15/group) were anesthetized with 87.5 mg/Kg ketamine and 12.5 mg/Kg xylazine and were treated with radiation at 8 Gy (Rad Source RS 2000, irradiator, Brentwood, TN, USA) given concurrently with 20 mg/m^2^ of cisplatin administered IP on days 13, 20, and 27. Shielding was performed such that the radiation field was only applied to the hind limb/tumor area. For combination anti-PD-1 experiments, DX400 or IgG1 isotype control antibody were administered at a dose of 200 micrograms IP every 4 days for a total of 5 doses on days 12, 16, 20, 24, and 28 in mice undergoing chemoradiation. Cisplatin and radiation dose were based on equivalent human dosing and prior experiments in this model [6].

Efficacy was determined based on tumor response over time and survival. Tumor dimensions were measured weekly using calipers and tumor volume calculated as volume = (width^2^)(depth). Animals were euthanized if the tumors reached 1.5 cm in any dimension, the animal became emaciated, or demonstrated functional leg impairment (criteria per approved animal protocol). When mice reached euthanasia criteria or a maximum of 140 days, survival analysis was performed, standardizing to the predefined 2500 mm^3^ tumor volume endpoint, as described previously [6]. Tumor growth curves represent average tumor volume of groups over time (Figure 1B(ii)) or individual growth curves (Figure 1B(iii,iv)).

### 2.2. Human Subjects

Peripheral blood specimens were collected from 35 patients with AJCC 7th edition stage III-IVA HNSCC treated on a clinical trial (NCT02586207) at Sanford Health, Sioux Falls, SD, which explored treatment with the monoclonal antibody (mAb) PD-1 inhibitor, pembrolizumab 200 mg every 3 weeks in combination with a low dose of cisplatin (40 mg/m^2^ 6 times weekly) and radiotherapy (70 Gy fractionated at 2 Gy once daily over 35 days). Clinical trial data reporting safety and efficacy were previously reported [30]. Treatment and relative sample collection time points are outlined in Figure 2A. Healthy controls were obtained from de-identified blood bank donors (demographic data unavailable).

### 2.3. Human Sample Collection and Processing

Ten to twenty mL of blood were collected from each patient at each time point. Peripheral blood mononuclear cells (PBMCs) were isolated by standard Ficoll-Hypaque density gradient centrifugation method. Cells were resuspended in 10% DMSO, 90% fetal bovine serum for storage. The plasma portion was further centrifuged at 800× *g* for 20 min, followed by collection of supernatants to ensure that cell-free plasma was obtained. PBMCs and plasma were cryopreserved until use.

### 2.4. Multicolor Flow Cytometry

Multiparameter flow cytometry was performed on frozen PBMCs, as previously described [31]. One vial of PBMCs from baseline, week 3, and week 21 were thawed per patient. PBMCs were counted and assessed for cell viability (10^7^ viable cells) with trypan blue exclusion. Median cell viability was 92% (89–95% interquartile range), and less than 5% of PBMC samples assayed from the trial (4/81) had viabilities <80%. Samples with viability <80% were annotated in the analysis to ensure that low viability did not impact the ability to detect PD-1, TIM-3 and LAG-3. PBMCs were stained with three different antibody panels (Appendix A). These panels identified markers of known immune checkpoint proteins PD-1, TIM3 and LAG-3 signaling (panel 1), cytotoxic (CD3+CD8+) T cells and helper (CD3+CD4+) T cells and regulatory T cells (Treg; CD4+CD25+CD127-FoxP3+) (panel 2) and B cells (CD19+) and MDSC-like cells (MDSC-like; CD33+CD11b+CD14+HLA-DRlo/-CD15-) and (panel 3). Following staining, samples were analyzed on a BD FACS LSRFortessa SORP (BD Biosciences, San Jose, CA, USA) equipped with five lasers (UV 355 nm, Blue 488 nm, Yellow Green 561 nm, Violet 405 nm and Red laser 640 nm). The cytometer is calibrated daily before use with cytometer setup and tracking beads to ensure quality performance. The fluorochromes used are shown in Appendix A. Voltages were adjusted to allow both negative and positive signals to be visualized and to minimize spectral overlap between channels. FCS files were exported and analyzed using FlowJo v9.7 (Treestar, Ashland, OR, USA). Gating strategies with non-viable cells excluded and negative gates set based on fluorescence minus one controls are provided in Appendix A. Compensation was performed in FlowJo using beads (BD Biosciences) that were single-stained with each of the antibodies in the 3 panels. The frequency of all subsets was calculated as a percentage of PBMC to help eliminate the bias that could occur in the smaller populations with fluctuations in leukocyte subpopulations.

### 2.5. T-Cell Clonality

To study T-cell receptor rearrangement and clonality, we selected 14 representative patients (7 with HPV+ disease and 7 with HPV− disease) for further analysis. Genomic DNA (gDNA) was isolated from PBMCs at baseline and during treatment (week 6) and compared with gDNA isolated from initial tumor biopsies for each cancer patient using a DNeasy Blood & Tissue kit (Qiagen, Hilden, Germany). Further analysis was performed using a prequalified multiplex polymerase chain reaction (PCR) assay, composed of forward and reverse primers that directly target the family of variable (V) genes (forward primers) and joining (J) genes (reverse primers). Each V and J gene primers act as priming pairs to amplify somatically recombined TCRs, and each primer contained a specific universal DNA sequence. Following the initial PCR amplification, each amplicon was amplified a second time with forward and reverse primers containing the universal sequence and adaptor sequence needed for DNA sequencing (immunoSEQ RUO Kit Adaptive Biotech). Immunosequencing data were analyzed using immunoSEQ Analyzer software (Adaptive Biotechnologies, Seattle, WA, USA).

### 2.6. Statistical Analysis

For most figures, aggregated data were presented using mean values to represent the central tendency, and standard error mean (SEM) to represent variability. Two-tailed paired samples Wilcoxon test or unpaired t-test were used to determine significance of differences. *p* ≤ 0.05 was used as the cutoff for significance. Statistical analysis for the survival graphs was performed using the log-rank test with α = 0.01. Sample sizes for the animal experiments were based on prior experience with the animal model using other immune therapies alone and in combination with radiation therapy, while maintaining a focus on limiting the animals sacrificed. Please see the Appendix A for details of the calculations performed for sample size. No formal sample size for human biospecimen experiments was utilized, as the sample size was determined by the endpoints from the clinical trial.

## 3. Results

### 3.1. PD-1 Blockade Synergizes with Chemoradiotherapy in an HPV+ HNSCC Syngeneic Model

Flow cytometry studies revealed that untreated HPV+ mEERL tumors harbor a clear decrease in the percentage of CD45+ immune cells over time (Figure 1A). Despite an initial rise at 14 days post-tumor implantation, cytotoxic (CD3+CD8+), helper (CD3+CD4+), and Treg (CD3+CD4+FOXP3+) T-cell populations decreased by day 28 with a concordant increase in the percentage of granulocytic MDSC populations. During this time, the percentage of CD4+ and CD8+ cells with proliferative capacity (KI67+) clearly decrease in the tumor microenvironment. Evaluating relevant exhaustive immune checkpoint expression (PD-1, TIM-3, and LAG-3) on CD4+ and CD8+ T cells demonstrated that, while PD-1 and LAG-3 expression peak on day 14 and then decline, PD-1 persists at higher than baseline expression by day 28. These findings indicate a tumor microenvironment with more suppressive immune cell populations (MDSCs) and an exhausted T-cell immunophenotype. Several of these features have been identified as characteristics of “cold tumors” that are less responsive to anti-PD-1 therapies [32].

As T-cell PD-1 expression was a persistent finding in the tumor microenvironment, we evaluated the impact of anti-PD-1 therapy. As outlined in Figure 1B, initial experiments evaluating anti-PD-1 monotherapy showed no activity in terms of tumor response or survival compared to the isotype control. Based on findings supporting chemoradiotherapy as a modality that augments immune response [33,34], we evaluated the addition of PD-1 blockade to this treatment. The combination of anti-PD-1 therapy with chemoradiotherapy demonstrated improved tumor clearance and a significant improvement in survival as compared to isotype control-treated mice (58% versus 8%, *p* = 0.004) at day 120.

### 3.2. Circulating T-Cell Populations Decline with an Apparent Rise in Immunosuppressive Monocytes during Chemoradiotherapy despite PD-1 Blockade in HNSCC Patients

While characterization of the tumor-infiltrating immunocytes is feasible in a murine model, determining the flux in immune response to human tumor under treatment is more challenging and prompted an evaluation of circulating immunocytes instead. Prior reports demonstrate that cisplatin-based chemoradiotherapy results in a profound decrease in circulating T-cell populations and a relative increase in CD33b+CD11b+HLADRlo monocytes (representative of MDSC-like populations) and Treg populations during HNSCC treatment [17]. We sought to evaluate the effect of adding PD-1 blockade on circulating immunocytes with a similar chemoradiotherapy approach in human subjects with HNSCC treated on a clinical trial. Peripheral blood specimens from 21 HPV+ and 14 HPV− patients with stage III and IVA HSNCC were evaluated at baseline, during concurrent treatment, and after treatment completion as defined in Figure 2A. The demographics of the patient population are outlined in Table 1.

The majority of cases were male with Stage IVA HPV+ oropharynx cancers. First, we investigated the balance between effector T cells (CD8+ or CD4+) and suppressor immune cells (M-MDSCs and Tregs) at baseline between the HNSCC population versus similar age-matched healthy donors. HNSCC patients had significantly increased circulating MDSC-like and Treg populations with decreased CD19+ B cells. However, there were no significant changes in CD4+ and CD8+ T cells compared with healthy controls (Figure 2B). These data suggest that the balance between effector and suppressive peripheral immunocytes in HNSCC patients are skewed towards a suppressive circulating immunophenotype even before the initiation of therapy.

We evaluated the change in immunocyte populations during therapy and discovered that CD4+ T-cell levels decrease significantly with no significant changes on CD8+ T cells during and following treatment (Figure 3A). In parallel, the percentage of Treg cells remained stable throughout therapy, but MDSC-like populations strikingly increased (more than 1.5-fold), remaining high even 1 year post-treatment (Figure 3A). Additionally, we identified a significant decrease in the ratio CD4+/CD8+ and a sharp reduction in the CD8+/MDSC-like effector/suppressive balance that persisted for 1 year after treatment. B-cell populations, however, recovered at 1 year post-treatment to proportions similar to healthy controls (Figure 2B and Figure 3A). Contrary to findings previously reported with chemoradiotherapy alone [17], we found no significant changes in the CD8+/Treg ratio (Figure 3B) at any time point. Taken together, our data suggest that the overall effect of chemoradiotherapy with PD-1 blockade attenuates the decline in circulating T-cell populations observed during therapy; however, MDSC-like cell levels still rose significantly and persisted up to a year after therapy completion. While there was some inter-patient variation, there was no clear signal that correlated with HPV status, treatment response or disease recurrence.

### 3.3. During Chemoradiotherapy, PD-1 Blockade Reduces PD-1 Expressing T-Cell Populations, However, with a Concordant Rise in Other Exhaustive Checkpoint Expression

As previously mentioned, prior research demonstrates that the percentage of peripheral PD-1-expressing CD4+ T cells rises during chemoradiotherapy treatment [17], which, in turn, may contribute to immune escape. To determine the pharmacodynamic effect of PD-1 blockade during this treatment, we evaluated the percentage of PD-1 expressing CD4+ and CD8+ total and effector memory (EM) T cells (CD45RA−CCR7−). We found that the percentage of PD-1 expressing CD4+ and CD8+ positive cells significantly decreases after 3 weeks of treatment and remains lower than baseline while on active therapy. By 1 year after therapy completion, this cell population returns to baseline or rises above pretreatment levels (Figure 4A,B). To determine if other exhaustive checkpoint markers are also altered, we evaluated TIM-3 and LAG-3 expression on CD4+ and CD8+ total and effector memory (EM) T cells. In contrast to the drop in PD-1 expression that occurs during treatment, CD4+ and CD8+ T cells expressing TIM-3 and LAG-3 largely increase during treatment (Figure 4C–F), with the exception of LAG-3 expressing CD8+ T cells, which showed no significant change. Furthermore, no differences based on HPV status or treatment response were found in the percentage of PD-1-, TIM-3- and LAG-3-positive cells across populations (data not shown). These data suggest that anti-PD-1 therapy results in a decline in circulating PD-1 expressing T-cell populations with a compensatory rise in other populations expressing other exhaustive immune checkpoints.

### 3.4. Chemoradiotherapy in Combination with PD-1 Blockade Increases Clonal Selection in PBMC T-Cell Repertoire

Since the induction of adaptive immunity has been observed during chemoradiotherapy for several types of cancer including HNSCC [36], we decided to explore this in HPV+ and HPV− HSNCC patients treated in our clinical trial. Specimens were evaluated at baseline (tumor and blood) and 6 weeks after (blood) starting therapy with anti-PD-1 treatment and chemoradiotherapy; there was a clear enrichment in clonality throughout treatment and a concordant decrease in the number of productive rearrangements. Additionally, we found a clear decrease in fraction of circulating B- and T-cell populations during the treatment in a similar manner compared with Figure 2B (Figure 5A). The reasons for the prognostic differences between HPV+ and HPV− in HNSCC patients have been under investigation [4]. Therefore, we analyzed our data for differences of TCR clonality and enrichment in these patients, as well as a comparison between baseline and treatment TCRs from peripheral blood. Interestingly, HPV+ and HPV− HNSCC patients demonstrated comparable clonality frequencies for PBMCs at baseline or during treatment at week 6. In PBMCs, the HPV+ and HPV− patients manifested no significant difference in TCR richness at both baseline and during treatment (Figure 5B). Despite some heterogeneity in clonality across the participants, there was no correlation between treatment failure.

Given emerging data on the contribution of tumor neoantigens to the immune response, we evaluated tumor and concordant PBMC specimens at baseline and during treatment to explore dominant clones. While chemoradiotherapy significantly decreases the number of productive rearrangements, we found a clear decrease in B- and T-cell fractions during the treatment. This correlates with our flow cytometry findings in Figure 2B (Figure 5A). As previously mentioned, HPV+ and HPV− HNSCC have unique prognostic and molecular profiles. As such, we compared these findings in HPV+ and HPV− patients. As outlined in Figure 5C, a number of expanded and contracted clones occurred for each patient during treatment that were present in initial tumor biopsies. While all patients demonstrated changes in TCR repertoire (average of 193 expanded clones and 29 contracted clones per sample), expanded/contracted TCR clones were different between patients, suggesting unique clonal changes. Of those with expanded clones during treatment, 17% of the top-30 expanded populations were present in the tumor at baseline. Though the antigen-specific targets of these clones are unknown, they could represent neoantigen-specific clones. Of the other top-30 expanded TCR rearrangements not present in initial tumor sample tissue, many had previously been reported in the literature (red sequences) and may represent common TCR sequences found across patients. Regarding TCR contracted sequences, 60% were initially present in the tumor biopsy and similar to expanded repertoire, some of them are already published (Figure 5C). To summarize, multimodality treatment with PD-1 blockade plus chemoradiotherapy induces a clear clonal expansion, selecting unique and shared TCR sequences.

## 4. Discussion

In this study, we investigated the preclinical and clinical effect of PD-1 blockade during chemoradiotherapy for HNSCC. Our preclinical findings demonstrate an improvement in efficacy by the addition of PD-1 blockade to chemoradiotherapy. This response contrasted with the lack of single-agent activity with anti-PD-1 monotherapy in this model. We took this approach into the clinic and added PD-1 blockade to standard chemoradiotherapy in HNSCC. Data from on-treatment blood specimens demonstrated a pharmacodynamic effect of anti-PD-1 therapy on circulating T-cell populations with knockdown of PD-1 expressing populations during treatment. This is the opposite of what has been seen during standard therapy, where PD-1 expressing population rise significantly during treatment [17]. Finally, TCR data demonstrate clonal expansion of potential neoantigen-specific T-cell clones during combination therapy. These major observations help characterize the peripheral blood immune response during multimodality therapy with PD-1 blockade in HNSCC.

Findings from our syngeneic model suggest that monotherapy with PD-1 blockade may not be effective, which is the case for many malignancies. This mirrors clinical practice, where only approximately 10–20% of patients with recurrent/metastatic HNSCC respond to PD-1 blockade with the approved agents pembrolizumab [37,38] and nivolumab [39]. Our model harbors infiltrating immune cell populations similar to a low or non-inflamed immunophenotype [32]. Data in HNSCC and other tumor types show that both chemotherapy and radiation independently induce tumor inflammation through recruitment of tumor-infiltrating lymphocytes [40] and increased antigen presentation [41,42]. Previous findings with the mEERL mouse model show a clear need for an intact immune response for tumor clearance during chemoradiotherapy [6]. While chemoradiotherapy may induce an inflamed phenotype and set the stage for immune clearance, the PD-1:L1/2 interaction can still contribute to immune escape during this treatment approach. Our current work demonstrates that this interaction can be overcome by the addition of an anti-PD-1 mAb in this murine model.

Consistent with findings in the mEERL model, changes in the circulating immunocytes from the clinical trial suggest that PD-1 blockade also occurs in humans during concurrent chemoradiotherapy in the context of PD-1 inhibitor. Similar to other data in this setting [17], overall T-cell populations declined during therapy and MDSC-like populations rose consistent with a suppressive immunophenotype. However, contrary to these data, we demonstrate a clear knockdown of circulating PD-1 expressing CD4+ and CD8+ lymphocytes. This supports a pharmacodynamic effect of anti-PD-1 therapy on PD-1 expressing peripheral lymphocytes and suggests that at least circulating T cells appear to demonstrate some shift away from this exhaustive phenotype; however, compensatory mechanisms of exhaustion may still persist. Additionally, TCR data support overall expansion of T-cell clones during therapy, some of which were present in the tumor microenvironment at baseline, thus suggesting a possible expansion of neo-antigen-specific T cell populations during treatment.

Unanticipated results from the peripheral blood studies raise concern for ongoing immune exhaustion during this treatment. First, MDSC-like populations still rose during treatment similar to what has been seen with standard chemoradiotherapy [17]. Interestingly, these populations remained elevated even at one year post-treatment. While these populations have been implicated in preventing autoimmune and inflammatory toxicity [43] in normal tissue, they could also create an avenue for immune escape in the post-treatment setting [44]. Further complicating these findings is the rise in LAG-3 and TIM-3 expressing T-cell populations during treatment. Similar findings have been seen in HNSCC patients treated with anti-PD-1 monotherapy, where TIM-3 expression concordantly rose with PD-1 blockade mediated by the PI3K-AKT pathway [45]. Synergy of dual blockade of PD-1 with either TIM-3 [45] or LAG3 [46] with blocking antibodies has been seen in preclinical settings. Recent clinical trial data also support dual blockade of LAG3 and PD-1 in melanoma [47]. However, this has been done in the absence of chemotherapy and radiation. As a result, the implications of our findings remain unclear. We did not see a correlation with disease recurrence or autoimmune toxicity in our clinical study, though this could be due to the limited number of clinical events in those analyzed. Despite the unknown implications, our findings are hypothesis-provoking. It is possible that these populations rise during this therapy to promote self-protection from local toxicity based on the inflammatory nature of chemoradiotherapy [42]. However, this could serve as a pathway for tumor immune escape [48]. As larger, randomized studies are now investigating this treatment approach in HNSCC [49,50], immune biomarker data will hopefully shed light on the impact of peripheral blood immune response during treatment. This is particularly important, as one of these trials did not meet its primary efficacy outcome [51]. With better understanding of T-cell dynamics during this combination therapy, it is possible that other combinations of novel immuno-oncology agents targeting TIM-3, LAG-3 or other checkpoints could be rationally integrated to enhance clinical benefit.

There are several limitations in our findings. First, while tumor response and survival data from the murine model demonstrate clear synergy with the addition of anti-PD-1 therapy to chemoradiotherapy, we do not have clear mechanistic data explaining this synergy. Similar to our clinical trial, we were not able to obtain adequate on-treatment specimens to extensively evaluate tumor-infiltrating lymphocyte populations during treatment. Without these data, what is happening in the tumor microenvironment during concomitant therapy remains undefined. Due to the limitations of mouse circulating volume, we also were not able to correlate circulating immunocyte population to the tumor-infiltrating population, which would have potentially shed light on the human immunocyte analysis. Additionally, we have the same limitation from our human specimens with the lack of on-treatment tumor biopsies, but the peripheral blood findings suggest a shift away from PD-1 expressing lymphocytes in the periphery during PD-1 blockade. We can only assume that this is due to blockade of this pathway during treatment. However, without tumor data, we remain uncertain as to the molecular interactions occurring in the tumor microenvironment. While these limitations are present, our data are still valuable as they provide the first analysis of the host immune response to the addition of anti-PD-1 therapy during chemoradiotherapy in this disease.

## 5. Conclusions

In conclusion, our preclinical and clinical findings provide insight into the use of anti-PD-1 therapy during concurrent chemoradiotherapy for HNSCC. Our murine model represents a model that displays clear therapeutic effect with this approach. Correlative translational data generated from our first in-human trial demonstrate a clear pharmacodynamic signal by adding pembrolizumab to chemoradiotherapy. However, additional findings suggest other mechanisms of immune exhaustion that may occur during this treatment, supporting the need for ongoing research to explore strategies to better understand and overcome these potential barriers. Ultimately, our goal is to use this model and translational data to move into clinical trials that rationally utilize novel immuno-oncology agents to improve therapy.

## Figures and Tables

**Figure 1 cancers-14-02499-f001:**
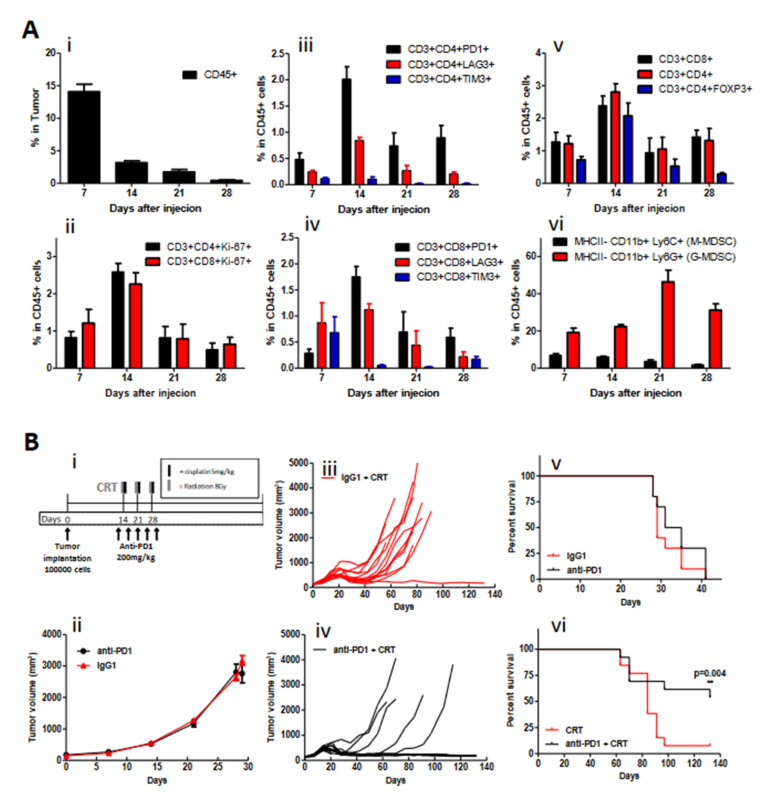
PD-1 Blockade during chemoradiotherapy enhances tumor clearance and survival in the mEERL model. (**A**) time course of changes in the relative number (expressed as percentage of CD45+ immune cells) (**i**) of CD3+ CD4+ and CD3+ CD8+ T cells (proliferative Ki67 cells (**ii**)) with PD-1, TIM-3, LAG-3 expression (**iii**,**iv**), Treg (FoxP3+) cells (**v**), and suppressor (M-MDSC and G-MDSC) immunocytes after mEERL tumor implantation (**vi**). (**B**) Experimental treatment schema (**i**), Average tumor volume and survival ratio after treatment alone with IgG1 isotype control and PD-1 inhibitor (**ii**) and individual growth curves in combination with chemoradiotherapy (CRT) (**iii**,**iv**) in mEERL murine model. Survival ratio with anti-PD-1 monotherapy versus IgG1 isotype control (**v**) and combination CRT + IgG1 isotype control versus and anti-PD1 + CRT (**vi**) ** (*p* ≤ 0.004).

**Figure 2 cancers-14-02499-f002:**
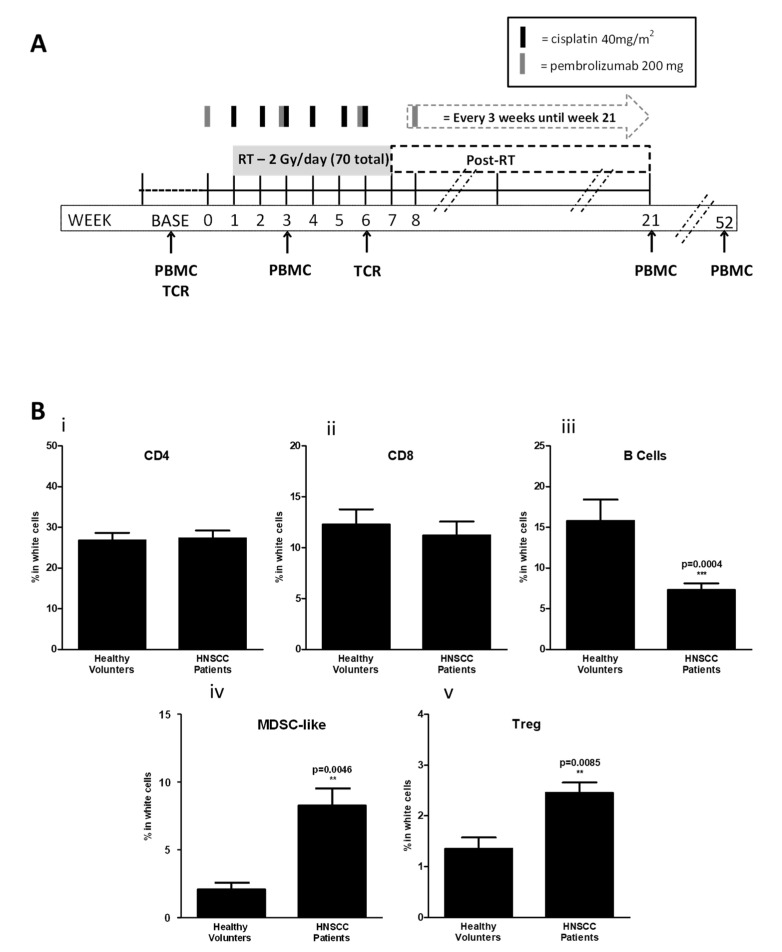
Biospecimen collection and comparison of immunocyte populations in healthy controls and HNSCC patients. (**A**) Outline of study treatments and relative PBMC collection. PBMCs were evaluated at baseline (BASE) prior to treatment and at multiple time points (weeks 3, 6, 21 (3-month), and 52 (1-year)) during and after the completion of concurrent therapy with multicolor flow cytometry (PBMCs) and T-cell receptor sequencing (TCR) studies performed at the time points indicated. (**B**) PBMCs were harvested from healthy volunteers (*N* = 10) or study patients (*N* = 35) from baseline draw and flow cytometry performed to determine the relative number of circulating CD4 T cells (**i**), CD8 T cells (**ii**), B cells (**iii**), MDSC-like cells (**iv**) and regulatory T cells (Treg) (**v**). Cancer patients had a significant increase in circulating M-MDSC and Treg, and significant decrease in B cells compared to healthy controls. ** (*p* ≤ 0.01) and *** (*p* ≤ 0.001).

**Figure 3 cancers-14-02499-f003:**
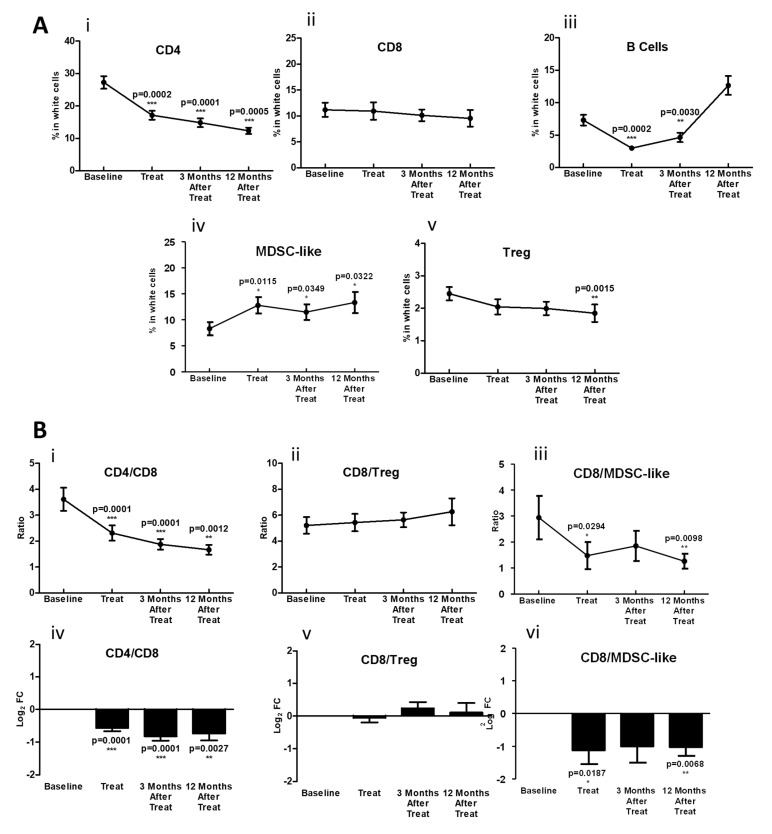
Effect of PD-1 blockade during chemoradiotherapy on circulating immunocytes. (**A**) time course of changes in the relative number (expressed as percentage of live white cells) of effector (CD4+ (**i**) and CD8+ (**ii**)) T cells, B cells (**iii**) and suppressor (MDSC-like (**iv**) and Treg cells (**v**)) before and after treatment. (**B**), CD4+/CD8+ (**i**), CD8+/Treg (**ii**) and CD8+/MDSC-like cells (**iii**) and expressed as the Log2 fold change (Log2 FC) of these ratios with respect to baseline (**iv**–**vi**). * (*p* ≤ 0.05), ** (*p* ≤ 0.01) and *** (*p* ≤ 0.001) with respect to baseline.

**Figure 4 cancers-14-02499-f004:**
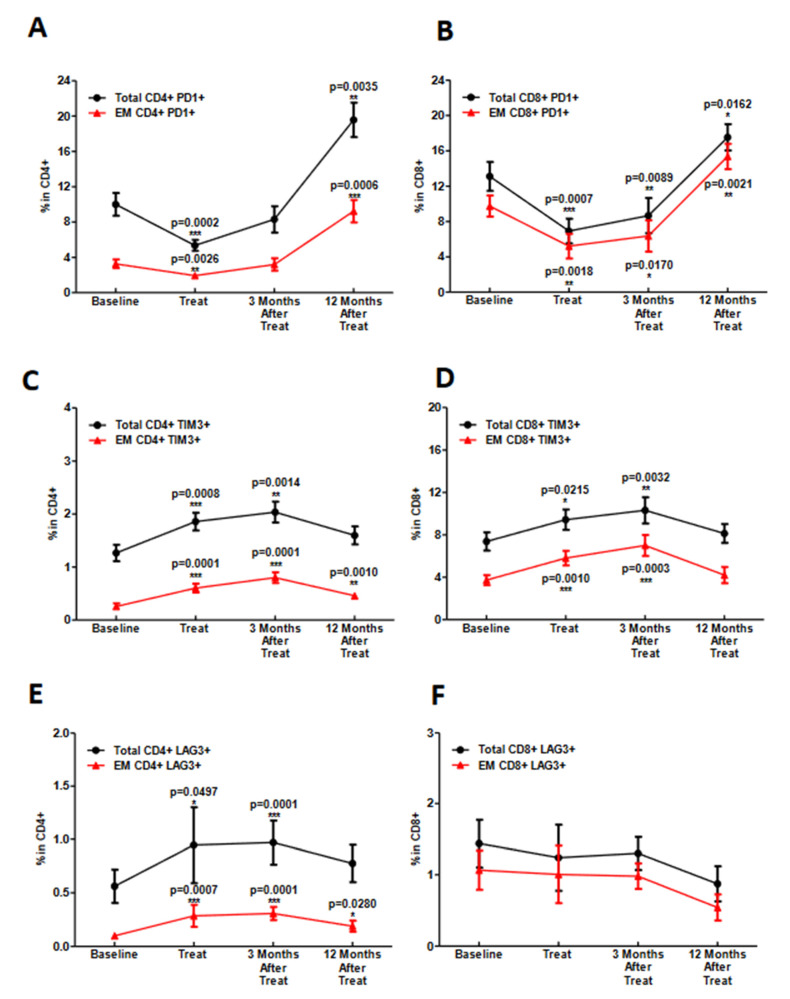
Effect of PD-1 blockade during chemoradiotherapy on PD1, TIM3 and LAG3 expression on circulating T cells. (**A**,**B**) time course of changes in the relative number of PD-1+ CD4+ and CD8+ T cells expressed as percentage of total CD3+CD4+, CD3+C8+ and EM T cells (CD3+CD4+ CD45RA−CCR7−, CD3+CD8+ CD45RA−CCR7−). (**C,D**) time course of changes in the relative number- of TIM3+ CD4+ and CD8+ expressed as percentage of total CD3+CD4+, CD3+C8+ and EM T cells. (**E**,**F**) time course of changes in the relative number expressed as percentage of total CD3+CD4+, CD3+C8+ and EM T cells of LAG3+ CD4+ and CD8+ T cells. * (*p* ≤ 0.05), ** (*p* ≤ 0.01) and *** (*p* ≤ 0.001) with respect to baseline.

**Figure 5 cancers-14-02499-f005:**
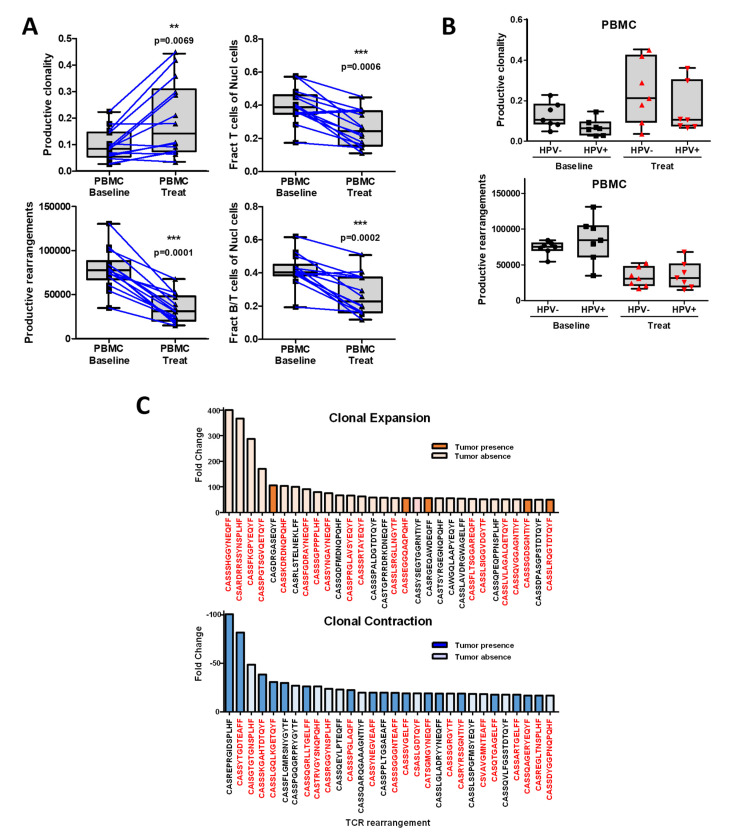
Multimodality treatment in combination with PD1 inhibition increases clonal selection in PBMC T-cell repertoire. (**A**) Changes of clonality in the PBMC samples before and after treatment. Treatment with pembrolizumab plus chemoradiotherapy decreases the number of productive rearrangements in PBMCs. Fraction T cells and B/T cells of nuclear cells drops during treatment. ** (*p* ≤ 0.01) and *** (*p* ≤ 0.001) with respect to baseline. (**B**) HPV+ and HPV− demonstrate comparable clonality frequencies for PBMCs at baseline or during treatment. In PBMCs, the HPV+ and HPV− patients manifest no significant difference in productive rearrangement at baseline and during treatment. (**C**) Top 30 TCR rearrangements (present in the tumor at baseline or absent at baseline) for both clonal expansion/contraction in all the PBMCs patient cohorts based on fold change. In red, TCR sequences that are already published.

**Table 1 cancers-14-02499-t001:** Patient and tumor characteristics from clinical trial adding pembrolizumab to chemoradiotherapy (NCT02255097).

Characteristic	Total Patients
	(*N* = 35)
Median Age, Years	61.6
Range	(38–81)
Sex	
Male	30 (85.7%)
Female	5 (14.3%)
Race	
White, Non-Hispanic	35 (100%)
Primary Site	
Oropharynx	22 (62.8%)
Larynx	9 (25.7%)
Hypopharynx	4 (11.5%)
TNM Stage (AJCC 7th ed.) [35]	
III	8 (22.8%)
IVA	27 (77.2%)
T0–1 *	8 (22.8%)
T2	11 (31.4%)
T3	11 (31.4%)
T4	5 (14.4%)
N0	3 (8.6%)
N1	4 (11.4%)
N2	28 (80.0%)
p16 (HPV) Status	
Positive	21 (60.0%)
Negative	14 (40.0%)

* One patient with unknown primary HNSCC.

## Data Availability

The data presented in this study are available on request from the corresponding author. The data are not publicly available due to patient privacy.

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
