# Peer review of "Characterization of the Immune Response to PD-1 Blockade during Chemoradiotherapy for Head and Neck Squamous Cell Carcinoma"

_cancers, 2022, doi:10.3390/cancers14102499_

Round 1
Reviewer 1 Report
This present article by Callejas-Valera et al helps to understand the treatment failure in chemoradiotherapy for HNSCC and also highlights potential competing mechanisms of immune evasion. Moreover, they characterized tumor infiltrating immune cell populations to define the extent of PD-1 + T-cell infiltration. In addition to that their TCR data demonstrates clonal expansion of 457 potential neoantigen specific T-cell clones during combinational approach. These observations would further help in characterizing the peripheral blood immune response during multimodality therapy with PD-1 blockade in HNSCC. I am in principle supportive of accepting this work for publication. However, I have few suggestions to improve the manuscript for publication.
Minor comments:
- Table 1 missed the alignment, I would like them to correct Table-1
- Minor spelling mistake detected in some places, for instance C57BJ/6 misspelled as C57BlJ/6
Author Response
Comment 1: Table 1 missed the alignment, I would like them to correct Table-1
Author Response: Thank you for noticing this error. This has now been edited to align with the margins
Comment 2: Minor spelling mistake detected in some places, for instance C57BJ/6 misspelled as C57BlJ/6
Author Response: Again, thank you for noticing this spelling error. The manuscript was proofread again by our team. This spelling error and others were changed. Please see redlined version for details and each correction.
Reviewer 2 Report
This manuscript examines preclinical and clinical models exploring the interaction between PD-1/L1 blockade and chemoradiation for the treatment of HNSCC. This is an important area of research that is needed to find better ways to incorporate immunotherapy into existing treatment paradigms for HNSCC patients to improve outcomes. The authors provide valuable data regarding the host immune response to immunotherapy in this population that should help drive future studies.
This manuscript is well written and provides strong support for its conclusions. I have no further edits or suggestions.
Author Response
Comment: This manuscript is well written and provides strong support for its conclusions. I have no further edits or suggestions.
Author Response: Thank you for your review and encouraging comments. Please see the edited version highlighting minor revisions requested by the other reviewers and let us know if these changes are acceptable
Reviewer 3 Report
Sir, I have reviewed the manuscript "Characterization of the Immune Response to PD-1 Blockade During Chemoradiotherapy for Head and Neck Squamous Cell Carcinoma" submitted by Juan Luis Callejas-Valera and co-workers to Cancers recently.
The authors focused on an important issue of HNSCC and the potential of immunotherapy (anti PD/PD-L1) to improve therapeutic outcomes.
The authors have presented a very nice set of data and nicely illustrated the production of immunologically "cold tumours” that are less responsive to anti-PD-1 therapies. The murine data are in a somewhat small scale also correlated to human patients.
I believe that it is convincing that the balance between effector and suppressive peripheral immunocytes in HNSCC patients is skewed towards suppressive phenotypes. Sadly, these data suggested that anti-PD-1 therapy results in a decline in circulating PD-1 expressing T-cell populations with a compensatory rise in other immunocytes expressing other exhaustive immune checkpoints.
It is nothing surprising that data from this model suggest that monotherapy with PD-1 blockade may not be effective. Indeed, this is what we see routinely in daily practice in oncology. However, there is also something positive and well documented and supported by this study. Data in various tumours including HNSCC suggested earlier that localised treatment, namely chemotherapy, cryotherapy and radiation independently induce tumour inflammation by immunocytes. Of note, it increases the recruitment of tumour infiltrating lymphocytes to "cold tumours" and increases antigen presentation.
It raises a very provocative question regarding the potential abscopal effect in chemo/radio/immunotherapy treated group. I believe this aspect is worthy of attention and at least a brief comment in discussion. Have theauthors observed anything consistent with the definition of the abscopal effect? There are sporadic case-reports of clinical cases where radiotherapy induced regression even of nontarget tumours (or induced complete remission). More recently, there are publications describing this clinically attractive phenomenon on the background of antiPD immunotherapy (in combination with radio- or more recently cryotherapy). I believe that the presented data are very well fitting into the context and presented immunological phenomena worthy of clinical attention. The need for combined therapy is indeed a very great virtue of thepresented manuscript in my eyes.
Minor points:
a) I believe it would be good to add a concise paragraph on markers (exhaustive immune checkpoints) TIM-3 and LAG-3 to the Introduction section.
b) line 119-120 - I believe the tumour cell injection is described inadequately - please, afd details. This is a critical aspect. E.g. number of cells (10exp5 seems to be quite low, please verify this), volume?, medium or matrigel was used? Single/several site/s per animal? Implantation success rate?
c) I believe that the authors should be encouraged to provide a graphical abstract or an explanatory scheme to their conclusions. I believe it can further increase attractivity of the manuscript.
d) in animal related study, it is highly advisable (from ethical reasons) to reveal (e.g. as a supplementary material) also pre-experimental calculation of necessary group size and statistical approach to experimental design and test applied etc. It usually leaves a very impression impression that only necessary number of animals was sacrificed. Please, consider this suggestion.
Conclusion: as it is now, the manuscript represents a very convincing and well performed study. It is worthy of attention, publication and reading. I believe that authors can provide some minor changes as soon as possible to perfect their outstanding work.
Author Response
Question: It raises a very provocative question regarding the potential abscopal effect in chemo/radio/immunotherapy treated group. I believe this aspect is worthy of attention and at least a brief comment in discussion. Have the authors observed anything consistent with the definition of the abscopal effect? There are sporadic case-reports of clinical cases where radiotherapy induced regression even of nontarget tumours (or induced complete remission). More recently, there are publications describing this clinically attractive phenomenon on the background of antiPD immunotherapy (in combination with radio- or more recently cryotherapy). I believe that the presented data are very well fitting into the context and presented immunological phenomena worthy of clinical attention. The need for combined therapy is indeed a very great virtue of the presented manuscript in my eyes.
Author Response: Thank you for the thorough review of our manuscript. In response to your comment regarding abscopal effect, we have not seen any evidence of this in our model. Our model behaves aggressively leading to distant metastases and death rapidly if the primary tumor site is not eliminated quickly, so evaluation of metastatic and relevant abscopal response has been challenging.
Comment a: I believe it would be good to add a concise paragraph on markers (exhaustive immune checkpoints) TIM-3 and LAG-3 to the Introduction section.
Author Response: We agree with this comment and have added a paragraph summarizing the importance of TIM-3 and LAG-3 as emerging checkpoints of interest. Additionally, we have included references pertaining to this new added background.
Comment b: line 119-120 - I believe the tumour cell injection is described inadequately - please, add details. This is a critical aspect. E.g. number of cells (10exp5 seems to be quite low, please verify this), volume?, medium or matrigel was used? Single/several site/s per animal? Implantation success rate?
Author response: We have added additional clarification on the points raised. The cell number is correct (10exp5). We added information highlighting the use of 100 uL of DMEM for a single injection site. Our group has extensive experience with this model and an implantation success rate of 100% with this approach.
Comment c: I believe that the authors should be encouraged to provide a graphical abstract or an explanatory scheme to their conclusions. I believe it can further increase attractivity of the manuscript.
Author Response: Thank you for this suggestion. We have created a graphical abstract to summarize our experimental approach and key findings.
Comment d: in animal related study, it is highly advisable (from ethical reasons) to reveal (e.g. as a supplementary material) also pre-experimental calculation of necessary group size and statistical approach to experimental design and test applied etc. It usually leaves a very impression that only necessary number of animals was sacrificed. Please, consider this suggestion.
Author Response: In the supplementary material, we have added a description of sample size calculations used for our studies based on our Institutional Animal Care and Use Committee (IACUC) protocol. In manuscript section 2.6, we have added language to direct the reader to the supplementary methods for this information.